# Sphingomyelinases and Liver Diseases

**DOI:** 10.3390/biom10111497

**Published:** 2020-10-30

**Authors:** Naroa Insausti-Urkia, Estel Solsona-Vilarrasa, Carmen Garcia-Ruiz, Jose C. Fernandez-Checa

**Affiliations:** 1Department of Cell Death and Proliferation, Instituto de Investigaciones Biomédicas de Barcelona, Consejo Superior de Investigaciones Científicas, 08036 Barcelona, Spain; naroaiu@hotmail.com (N.I.-U.); estelsolsona@gmail.com (E.S.-V.); 2Liver Unit Hospital Clínic i Provincial, IDIBAPS, 08036 Barcelona, Spain; 3Centro de Investigación Biomédica en Red (CIBERehd), 28029 Madrid, Spain; 4Department of Biomedical Sciences, School of Medicine, University of Barcelona, 08036 Barcelona, Spain; 5Southern California Research Center for ALPD and Cirrhosis, Keck School of Medicine, USC, Los Angeles, CA 90033, USA

**Keywords:** ceramide, sphingomyelin, acidic sphingomyelinase, neutral sphingomyelinase, hepatocellular carcinoma, alcoholic and nonalcoholic steatohepatitis

## Abstract

Sphingolipids (SLs) are critical components of membrane bilayers that play a crucial role in their physico-chemical properties. Ceramide is the prototype and most studied SL due to its role as a second messenger in the regulation of multiple signaling pathways and cellular processes. Ceramide is a heterogeneous lipid entity determined by the length of the fatty acyl chain linked to its carbon backbone sphingosine, which can be generated either by de novo synthesis from serine and palmitoyl-CoA in the endoplasmic reticulum or via sphingomyelin (SM) hydrolysis by sphingomyelinases (SMases). Unlike de novo synthesis, SMase-induced SM hydrolysis represents a rapid and transient mechanism of ceramide generation in specific intracellular sites that accounts for the diverse biological effects of ceramide. Several SMases have been described at the molecular level, which exhibit different pH requirements for activity: neutral, acid or alkaline. Among the SMases, the neutral (NSMase) and acid (ASMase) are the best characterized for their contribution to signaling pathways and role in diverse pathologies, including liver diseases. As part of a Special Issue (Phospholipases: From Structure to Biological Function), the present invited review summarizes the physiological functions of NSMase and ASMase and their role in chronic and metabolic liver diseases, of which the most relevant is nonalcoholic steatohepatitis and its progression to hepatocellular carcinoma, due to the association with the obesity and type 2 diabetes epidemic. A better understanding of the regulation and role of SMases in liver pathology may offer the opportunity for novel treatments of liver diseases.

## 1. Introduction: Metabolism and Regulation of Sphingolipids

Sphingolipids (SLs) are a family of lipids ubiquitously present in all cells that contain a long-chain base called sphingosine. Among the best understood and characterized SLs are ceramides, a heterogeneous group of lipids featuring an acyl chain linked to the sphingosine via an amide bond. The molecular identity of each ceramide molecule is determined by the specific fatty acyl moiety, which encompasses short to very long fatty acids (C2–C34). Complex SLs consist of ceramide and various head groups covalently attached to the hydroxyl group of the sphingosine through an ester or a glycosidic bond. For instance, a phosphocholine molecule is found in sphingomyelin (SM), while either a simple sugar residue or more complex carbohydrates determine the head group of glycosphingolipids (GSLs), a complex family of SLs that include cerebrosides, globosides and gangliosides [1,2].

Following their discovery from brain extracts in the 19th century, SLs have been merely considered as structural components of biological membranes. Their distinctive association with cholesterol defines specific domains of membrane bilayers that exhibit unique physical properties, and are used as scaffolds for key signaling platforms involved in diverse cellular processes [2]. Recent evidence over the last decade indicated that SLs are involved in a plethora of cellular functions, including cell growth, cell death, inflammation, immune responses, cell adhesion and migration, angiogenesis, nutrient uptake and responses to stress stimuli and autophagy. The wide range of biological effects of SLs is related not only to their structural diversity but also to their subcellular distribution and mechanism of generation [3,4]. Ceramide is the prototype SL that has been the most intensively characterized due to its role as a second messenger in the regulation of metabolism and cell death pathways in response to stress, apoptotic triggers and chemotherapy [5,6,7,8]. Ceramide can arise from the endoplasmic reticulum (ER), where the molecular machinery for its de novo synthesis resides [9]. In addition, ceramide can also be generated from the hydrolysis of sphingomyelin (SM) either at the plasma membrane or in intracellular acidic compartments (endosomes/lysosomes) by the activation of sphingomyelinases (SMases) [10]. Once generated, ceramide can be converted into a variety of metabolites. The deacylation of ceramide by ceramidases (CDases) yields sphingosine, which can be phosphorylated by sphingosine kinase (SK) to sphingosine-1-phosphate (S1P), another SL with an important role in cell signaling (Figure 1), or reacylated back to ceramide in the so-called salvage pathway [11,12]. Besides, the trafficking of ceramides to the Golgi mediated by the ceramide transfer protein (CERT) fuels the synthesis of complex GSLs and SMs (Figure 1). While the basal levels of ceramides in healthy cells are low, in response to many deleterious stimuli causing stress, apoptosis and cell death, cells trigger a rapid and transient mechanism of ceramide generation in specific sites due to SMase activation in distinct intracellular compartments, predominantly in lysosomes and the plasma membrane, that activate particular signaling pathways. Given the role of ceramide in hepatocellular apoptosis and fibrosis (see below), and since SMases represent the predominant pathway for the sudden generation of ceramide, in the present review, we summarize the role of SMases in liver pathology, including predominant chronic liver diseases, such as alcoholic and nonalcoholic steatohepatitis (ASH/NASH) and their progression to hepatocellular carcinoma (HCC) [13,14,15].

## 2. Ceramide Generation: De Novo Synthesis and Sphingomyelin Hydrolysis by Sphingomyelinases

### 2.1. De Novo Synthesis

The de novo pathway of ceramide generation occurs in the ER (Figure 1). In this pathway, the amino acid serine is conjugated with palmitoyl-CoA in a step catalyzed by the rate-limiting enzyme serine palmitoyl transferase (SPT). The product of the reaction sphinganine is acylated by ceramide synthases (CerSs) to dihydroceramide. Subsequent dehydrogenation catalyzed by dihydroceramide desaturase (DES) generates ceramide. In addition, CerSs also catalyze the reacylation of sphingosine to ceramide in the salvage pathway. Six different CerSs have been identified [16,17], which exhibit tissue-specific expression and substrate selectivity, thereby providing the basis for the generation of singular ceramide species of variable acyl chains in particular tissues. For instance, the ceramide synthase CerC2 is widely expressed and of major importance in the liver and preferentially incorporates long-chain C20–C24 acyl residues to generate C20–C24 ceramides. CerS3 is preferentially expressed in the skin and catalyzes the acylation of very long acyl chains up to C34:0 to sphinganine. The ceramide synthase CerC5 specifically catalyzes the generation of C16 ceramide, while the ceramide synthase CerC6 shows a wide substrate selectivity, and it is involved in the generation of C14, C16 and C18 ceramides [16,18]. Importantly, ceramides with different acyl chain lengths are generated in specific physiological and pathophysiological contexts in a tissue- and cell-dependent fashion. Despite this defined specific profile of ceramide synthesized by the different CerSs, there are compensatory mechanisms that offset the absence of specific ceramide species. In this regard, an increase in a particular CerS may regulate a specific ceramide pool that may affect the integrity and function of individual cell compartments, such as lysosomes, the ER or mitochondria. For instance, CerS2 knockout mice exhibit a compensatory increase in the levels of C16 in the liver, which triggers hepatocyte apoptosis and proliferation, leading to hepatocellular hyperplasia [19]. These changes in ceramide homeostasis translate into increased rates of hepatocyte apoptosis, mitochondrial dysfunction and mitochondrial ROS generation, as well as proliferation, that progress to the widespread formation of nodules of regenerative hepatocellular hyperplasia in aged mice. Progressive hepatomegaly and noninvasive liver tumors are observed in 10-month-old CerS2^−/−^ mice [20]. An important factor that controls the de novo ceramide synthesis involves the availability of the substrate palmitoyl-CoA, which is required for sphinganine synthesis and whose level increases in obesity, metabolic syndrome and related disorders (e.g., NASH) [21,22]. Thus, the obesity-related increase in palmitoyl-CoA is expected to enhance ceramide synthesis.

The de novo synthesized ceramide can then be distributed to distinct intracellular compartments, such as the Golgi, where it acts as a source of SM or GSLs. Besides, SM trafficking to lysosomes or the plasma membrane can generate discrete ceramide species due to the local activation of SMases, which initiate specific signaling pathways that account for the diverse biological actions of ceramide.

### 2.2. Sphingomyelinases: Types and Function

The SMases comprise a family of enzymes that catalyze SM hydrolysis with different biochemical characteristics, yielding ceramide and phosphocholine. The SMases encompass three subclasses based on their optimal pH and subcellular localization: acid (ASMase), neutral (NSMase) and alkaline (Alk-SMase). While Alk-SMase is mainly localized in the gastrointestinal tract and to some extent in the liver, NSMase and ASMase are ubiquitous and account for the generation of ceramide in specific intracellular compartments, predominantly in the plasma membrane and lysosomes, respectively (Figure 2).

Two forms of ASMases are encoded by the gene Smpd1. The ASMase associated with the endosomal/lysosomal compartment hydrolyses lysosomal SM delivered by lipoproteins or through the endocytic pathways. On the other hand, secretory ASMase is found in the plasma and in the conditioned medium of stimulated cells, and has a complex pattern of glycosylation as well as a longer in vivo half-life [23,24,25]. Although the secretory ASMase form has been reported to be dependent on Zn^2+^ [24], recent findings describing the crystal structure of mammalian ASMase revealed an N-terminal saposin domain and a catalytic domain, which adopts a calcineurin-like fold with two Zn^2+^ ions [26]. Whether this accounts for the selective dependence of the secretory ASMase on Zn^2+^ remains to be established. The mechanisms involved in the generation of the secretory ASMase are not fully understood. While the trafficking of lysosomal ASMase to the Golgi and processing by S1P are thought to generate secretory ASMase, recent findings in the protozoan parasite *Trypanosoma cruzi* suggested that conventional lysosomes fuse with the plasma membrane in response to an increase in intracellular Ca^2+^, releasing their contents extracellularly, where the resultant exocytosed ASMase from lysosomes remodels the outer leaflet of the plasma membrane [27]. Further work would be required to examine if this mechanism contributes to the genesis of the mammalian secretory ASMase. The Smpd2 and Smpd3 genes encode NSMase-1 and NSMase-2, respectively, both of which are Mg^2+^-dependent but differ in their subcellular localization and role in signaling pathways. In mammalian cells, NSMase-1 is found in the ER and the Golgi apparatus, while NSMase-2 promotes SM hydrolysis on the cytosolic face of the plasma membrane as well as in multilamellar bodies and the nuclear envelope (Figure 2) [28,29,30,31]. NSMase-2 is regulated by the TNFα and IL-1β cytokines and mediates cellular responses to stress and inflammation [32,33,34,35]. By contrast, due to the subcellular location of NSMase-1 in the ER and Golgi, it is unlikely that this isoform plays a role in signaling pathways, in line with data suggesting a lack of influence of NSMase-1 overexpression in TNFα-induced signaling pathways [36] or Fas-induced apoptosis [37]. The Smpd4 gene encodes a novel form of NSMase, NSMase-3, which is mostly found in skeletal muscle and the heart but not in the liver. Lastly, a mitochondrion-specific Smpd5 has been recently identified, with the highest expression in the testis, pancreas and fat tissue (MA-NSMase) [38]. Moreover, Alk-SMase does not share any structural similarity with NSMases or ASMases, belongs to the ectonucleotide pyrophosphatase/phosphodiesterase (NPP) family, and therefore is also known as NPP7 and is encoded by the ENPP7 gene. Following transcription, Alk-SMase is anchored on the surface of cell membranes, and in addition, it can also be released from this location by bile salts and pancreatic trypsin. Moreover, Alk-SMase accumulates in the gallbladder, and its activity depends on bile salts [39,40] and is thought to be mainly involved in hydrolyzing dietary SM and stimulating cholesterol absorption [41].

## 3. Physiological and Signaling Function of Sphingomyelinases

As mentioned, SMase’s enzymatic activity leads to a rapid and transient release of ceramide in specific discrete intracellular sites depending on the type of SMase activated. In the following section, we will briefly describe the mechanisms and signaling pathways underlying the effects of NSMase and ASMase. Consistent with their sites of activation and requirements for optimal activity, the resultant ceramide species generated target different pathways, such as PKCδ or KSR for NSMase or MAT1A or cathepsins for ASMase (Figure 3).

### 3.1. NSMase

From the different mammalian NSMases characterized to date, NSMase-2 appears to be the predominant isoform involved in cell physiology and in the activation of different signaling pathways. NSMase-2 specifically hydrolyzes the phosphocholine headgroup from SM at the plasma membrane and does not exhibit any phospholipase C-type activity against phospholipids, such as phosphatidylcholine (PC) or lysophosphatidylcholine. As mentioned, the principal biochemical features of NSMase-2 are its requirement for a neutral pH and divalent cations, such as Mg^2+^, for optimal activity. In addition, phosphatidylserine (PS), as well as other anionic phospholipids, including phosphatidic acid or phosphatidylinositol, stimulates enzymatic activity, while unsaturated fatty acids have been shown to mimic this behavior only in vitro [35,42]. Besides anionic phospholipids, NSMase-2 is regulated by phosphorylation in conserved serine residues, consistent with its identification as a phosphoprotein. In this regard, protein phosphatase 2B was found to bind a PQIKIY motif between the N-terminus and the C-terminus to dephosphorylate NSMase-2 [43]. However, it is still unknown whether protein kinases, such as p38 or PKCδ, directly phosphorylate NSMase-2.

NSMase-2 has been characterized as a mediator of TNF-α signaling, which involves the formation of a pentacomplex containing TNFR-1, NSMase-2, EED, RACK1 and FAN that results in the regulation of neurological (synaptic plasticity and neuronal cytotoxicity), vascular (vasodilation and adhesion) and inflammatory effects. As a mediator of TNF-α’s biological effects, NSMase-2 has been shown to induce apoptosis that can be prevented by the overexpression of Bcl-xL [44]. Recent studies indicated that NSMase-2 cooperates with Bax and Bcl-2 to activate the mitochondrial-dependent apoptotic machinery [45]. Interestingly, emerging data implicate NSMase-2 as a component of exosomes whose release takes place through a non-canonical pathway independent of endosomal sorting complexes required for transport (ESCRT) proteins. This new role of NSMase-2 has been involved in cancer development in breast xenografts, as the blockade of the NSMase-2-mediated exosomal release from 4T1 xenografts reduced tumor growth and lung metastasis through alterations in endothelial function [46]. Additionally, involving the action of NSMase-2 via exosomal release, it has been shown that NSMase-2 plays an emerging role in Alzheimer´s disease. Primary astrocytes from murine cortices treated with Aβ25–35 or Aβ1–42 died in parallel with the production of ceramide and caspase-3 activation. The treatment of wild-type mice and a 5XFAD Alzheimer’s disease mouse model with GW4869, an NSMase-2 inhibitor, resulted in lower exosome accumulation in the brains of 5XFAD mice, resulting in lower concentrations of Aβ1–42 and reducing Alzheimer´s pathology [47]. Linked to its role in cancer development, NSMase-2 has been shown to regulate cell differentiation and growth arrest. MCF7 breast cancer cells overexpressing NSMase-2 have a similar growth phenotype to control cells, and upon serum starvation, NSMase-2 expression prevents the progression of the cell cycle, which is retained in the G0/G1 phase, compared to control cells overexpressing empty plasmid. Moreover, it has been shown that cell confluence upregulates NSMase-2 to arrest cells in G0/G1 with the hypophosphorylation of the retinoblastoma protein and induction of p21, while NSMase-2 downregulation prevents this phenotype [32]. Thus, these findings identify NSMase-2 as a potential target for the modulation of inflammation, cell growth and apoptosis, emerging as novel target in cancer development and neurodegeneration.

### 3.2. ASMase

The generation of ceramide via ASMase regulates multiple signaling pathways, which are central to metabolism, Ca^2+^ regulation, autophagy and lysosomal homeostasis, and hence, ASMase emerges as an important signaling molecule regulating diverse cellular processes.

#### 3.2.1. ASMase and ER Stress

ER stress is a condition in which there is an accumulation of misfolded proteins in the ER. The unfolded protein response (UPR) is a complex signaling network, which is designed to restore protein homeostasis by reducing protein synthesis and increasing protein folding. Three signaling proteins are initially activated in the UPR—inositol requiring 1 alpha (IRE1α), PKR-like ER kinase (PERK) and activating transcription factor 6a (ATF6)—and this signaling is followed by the activation of several downstream targets [48,49,50]. The master regulator of UPR activation is the glucose-regulated protein 78 (GRP78, also known as BiP), which in physiological conditions, binds to IRE1α, PERK and ATF6 and prevents their activation. Upon an accumulation of misfolded proteins, GRP78 is released, enabling IRE1α, PERK and ATF6 to trigger the UPR.

ASMase per se has been shown to trigger ER stress through ceramide production and a subsequent impact on Ca^2+^ signaling. The disruption of ER Ca^2+^ homeostasis induced by exogenous ASMase treatment in hepatocytes caused ER stress due to Ca^2+^ release to the cytosol triggered by ceramide, which affected the ability of the chaperone BiP to bind UPR transducers [51]. Although the detailed mechanism whereby ASMase-induced ceramide generation perturbs Ca^2+^ regulation in the ER is not fully understood, the role of ASMase in this effect is consistent with reports showing that an aberrant lipid composition in the ER triggers Ca^2+^ release through sarco-endoplasmic reticulum Ca^2+^-ATPase (SERCA) regulation [52,53,54]. In addition, whether an ASMase-mediated increase in ceramide impacts the ER lipid composition remains to be investigated.

#### 3.2.2. ASMase and Autophagy

Autophagy is a highly regulated and complex catabolic process that is involved in the degradation of damaged or dysfunctional cell components, such as organelles such as mitochondria or peroxysomes, protein aggregates, lipid droplets or inflammasomes, through the fusion of autophagosomes with lysosomes [55]. During this process, cytoplasmic materials are recruited into a double membrane structure (the phagophore), which distends to form an autophagosome, a spherical structure with double layer membranes. This structure further fuses with lysosomes, creating an autolysosome, where all the contained materials are degraded by lysosomal hydrolases, while monomers such as free fatty acids (FFA) and amino acids are recycled. A failure to “digest” altered organelles can contribute to sustained alterations in metabolism and homeostasis, leading to cell dysfunction. This is best illustrated in the case of the defective elimination of disrupted mitochondria, as their accumulation can cause the release of stimulated ROS, disruption of lipid intermediate signaling and stimulation of proinflammatory cytokines.

Autophagy can be divided into several subtypes [56,57]. Macroautophagy is a protective non-selective mechanism that is activated during scarce nutrient availability with the aim of degrading cellular components in order to enhance the energy supply. By contrast, selective autophagy is a well-orchestrated process involving the recruitment of the autophagic molecular machinery to digest specific targets through specific protein signaling pathways [58]. This specificity makes autophagy a very important degradation process for preventing the accumulation of dysfunctional organelles and cell waste. Lipophagy, one of the several subtypes of autophagy, performs the selective degradation of intracellular lipids, and it has been described to have an important role in lipid metabolism and hepatic steatosis [59]. Besides regulating lipid storage, lipophagy also controls cellular energy homeostasis by providing FA to mitochondria to fuel β-oxidation and ATP synthesis. Cytosolic lipases have been known to catabolize triglycerides (TG) and lipid droplets in hepatocytes.

Autophagy disruption mediates the progression of many liver diseases, such as ASH/NASH, in which defects in autophagy promote steatotic and fibrogenic mechanisms [55]. Recent evidence suggests a role for ASMase in autophagy-mediated liver injury. For instance, hepatocytes from ASMase^−/−^ mice exhibit impaired autophagic flux, reflected specifically in the accumulation of dysfunctional mitochondria and resistance to high-fat diet (HFD)-induced hepatic steatosis [60]. Whether the role of ASMase in autophagy is mediated via the regulation of lysosomal cholesterol homeostasis and its impact on the fusion of lysosomes with autophagosomes remains to be fully elucidated [61]. Overall, while some data suggest a role for ASMase in autophagy-mediated steatosis in ASH/NASH, the potential involvement of ASMase in fibrosis via autophagy is still unknown and requires further investigation.

#### 3.2.3. ASMase and Lysosomal Membrane Permeabilization

Lysosomes are specialized membrane-bound organelles that contain a variety of hydrolytic enzymes responsible for the digestion and removal of macromolecules and organelles. Furthermore, lysosomes also play a crucial role in cell death regulation, in which lysosomal membrane permeabilization (LMP) followed by the leakage of lysosomal content is enough for cell death initiation [62,63]. Complete LMP causes a massive release of lysosomal content into the cytosol, which increases cytosolic acidification and enhances hydrolytic damage in different cellular components [62,64]. For instance, cathepsins (Cts), the major class of lysosomal proteases, are targeted to other organelles such as mitochondria, where they induce the release of proapoptotic factors. In addition, the acidification of mitochondria triggers mitochondrial membrane depolarization and Ca^2+^ handling impairment, which can trigger the recruitment of Bax in mitochondria to also initiate apoptosis [65]. Moreover, the acidification of the cytosol allows for some of the lysosomal proteases, such as Cts, to maintain their enzymatic activity and induce the proteolytic degradation of key cellular proteins during apoptosis [66,67].

LMP is induced by several different stimuli, such as ROS, lipids such as saturated FA or sphingosine, as well as cell death mediators, such as Bax. As lipids can trigger LMP, this event has been reported as one of the molecular mechanisms involved in NASH progression. For instance, palmitic acid (PA), one of the most abundant saturated FA in Western diets, induces LMP in hepatocytes, which triggers CtsB leakage into the cytosol, mitochondrial dysfunction followed by cytochrome c release and, ultimately, cell death [68,69]. As with ER stress and autophagy, ASMase is also related to LMP in NASH. ASMase deficiency causes resistance to PA-induced lipotoxicity in primary mouse hepatocytes. Consistent with the resistance to PA-mediated lipotoxicity, hepatocytes from ASMase^−/−^ mice are also resistant to amphiphilic lysosomotropic detergent-induced cell death. The increased levels of cholesterol in lysosomes upon ASMase deficiency are involved in such protection, as decreasing the lysosomal cholesterol content reverses the resistance of ASMase^−/−^ hepatocyes to amphiphilic lysosomotropic detergent and PA-induced cell death [60]. Thus, a deficiency of ASMase raises lysosomal cholesterol, which results in reduced LMP and a resistance to PA-induced lipotoxicity.

## 4. SMases and Liver Diseases

The liver is an important organ for lipid metabolism since it is involved in fatty acid β-oxidation, ketone body generation, cholesterol metabolism, lipoprotein synthesis and phospholipid metabolism. Hepatocytes secrete up to 5% of the newly synthesized SLs in the form of VLDL, constituting an important source of SLs. Specifically, the hepatic SM content is 7–8 fold higher than in subcutaneous and intra-abdominal adipose tissues [70]. Such high levels of SM in the liver are due to the fact that the liver effectively absorbs choline-containing compounds from phospholipid digestion in the intestinal tract, which are subsequently used for PC and SM synthesis [71,72]. Apart from synthesizing SM, the liver is also involved in SM hydrolysis, as the hepatic ASMase activity is higher than that in most organs. SMases are important for liver physiology and SM and ceramide homeostasis, and defects in SMases, particularly ASMase, result in profound alterations in liver function and the accumulation of SM, as illustrated in Niemann–Pick type A/B diseases (NPA/NPB) (see below) [73]. A high-fat diet (HFD), endotoxins or hepatitis B virus infections and liver cancer also influence hepatic SM homeostasis [74,75,76,77].

Alterations of SM levels subsequent to changes in the activity of SMases have been associated with the onset and progression of chronic liver diseases, of which NASH is of particular prevalence in the world. Several studies employing animal models as well as human specimens have provided strong evidence for ASMase and NSMase (e.g., NSMase-2) in liver diseases [1]. While ASMase is abundant in the healthy liver, hepatic NSMase-2 activity is very low in physiological conditions. Nevertheless, NSMase-2 is regulated by antioxidants, and hence, it can be activated during oxidative stress and hepatic GSH depletion [1]. Both NSMase and ASMase can be activated by cytokines and proinflammatory conditions, contributing to the progression of chronic liver diseases [35,78]. In the following section, we will briefly describe the contribution of both SMases to major chronic liver diseases (Table 1).

### 4.1. Alcoholic and Non-Alcoholic Steatohepatitis

Fatty liver disease is a spectrum of disorders that begin with hepatic fat accumulation (steatosis) that can progress to cirrhosis and HCC. Steatohepatitis is an advanced stage of fatty liver disease characterized by liver steatosis, inflammation, fibrosis and hepatocellular death [79,80] and encompasses both ASH and NASH, which share common histological and biochemical features [81]. The prevalence of NASH is of significant relevance due to its association with obesity and the type 2 diabetes epidemic. Unfortunately, the current available therapy for NASH is limited due to our incomplete understanding of the underlying mechanisms. In this regard, recent data have disclosed a pivotal role for ASMase in both ASH and NASH. In the case of ASH, ASMase activity and ASMase mRNA levels have been shown to be upregulated in liver biopsies from patients with acute alcoholic hepatitis [51,82,83,84]. In line with these observations, ASMase activity has been recently linked to biomarkers and alcohol use and abuse phenotypes in patients and healthy controls [85]. On the other hand, the expression and activity of ASMase have been shown to be increased in liver and serum samples from patients with NASH [86,87]. In line with the reported role of ASMase-induced ceramide generation in mediating hepatocellular apoptosis and fibrosis, ASMase has emerged as a potential new target for treatment for both ASH and NASH [88].

ASMase mediates the stress response and triggers hepatocellular apoptosis in response to TNF and Fas-induced fulminant liver injury. ASMase is required in TNF-induced hepatocellular apoptosis, TNF plus D-(+)-galactosamine (TNF/Gal)-mediated fulminant liver failure and Fas-induced lethal hepatitis [89,90,91], in which the recruitment of mitochondria is involved through ganglioside GD3 generation and further apoptosis via reactive oxygen species (ROS) generation and nuclear factor kappa-light-chain-enhancer of activated B cells (NFκB) inhibition [92,93]. In addition to cell death regulation, ASMase^−/−^ mice are resistant to alcohol- or HFD-induced lipogenesis and macrosteatosis, and hence, the pharmacological inhibition of ASMase using amitriptyline or imipramine (tricyclic antidepressants) in wild-type mice (WT) blocked alcohol- and HFD-induced steatosis [30,51,60,84]. As described above, there is a strong link between lipogenesis and ER stress, and ASMase has been shown to be required for either alcohol- or HFD-induced ER stress by regulating ER Ca^2+^ homeostasis. Besides the proapoptotic and prosteatotic role of ASMase, ASMase promotes fibrosis. Fibrosis is a wound-healing response to diverse types of insults, in which chronic injury drives the activation of hepatic stellate cells (HSCs) involved in the remodeling of the extracellular matrix and deposition of collagen, resulting in tissue scarring. Recent data have shown that ASMase is activated during HSC activation and required for HSCs’ transdifferentiation to myofibroblast-like cells that promote fibrogenesis [87]. In this event, ASMase regulates CtsB and CtsD, which are necessary for HSC activation and fibrosis initiation. Consistent with these findings, the pharmacological inhibition of ASMase using amitriptyline reduces hepatic fibrosis in a well-established CCl_4_-mediated-fibrosis model [94].

Autophagy and ER stress are mutually regulated, as defective autophagy induces ER stress [95]. In fact, ASMase is required for autophagy suppression-mediated ER stress in primary mouse hepatocytes [60]. Therefore, the induction of lipogenic genes through impaired-autophagy-mediated ER stress may be a significant pathway in the regulation of hepatic steatosis, in which ASMase has a central regulatory role. On the other hand, ER stress can also induce autophagy and further HSC activation and fibrosis [96]. The most characteristic feature of HSCs in physiological conditions is the presence of perinuclear membrane-bound droplets filled with retinyl esters. Thus, an increased autophagic flux may contribute to the loss of these lipid droplets during HSC activation [97,98]. Moreover, the genetic or pharmacologic inhibition of autophagy prevented HSC activation and fibrogenesis. Interestingly, the specific deletion of Atg7 in HSCs decreased fibrosis following sustained liver injury [99]. Overall, autophagy enhances energy production through the release of FFA from retinyl esters, which are used by HSsC as an energy supply for their activation and following fibrogenesis. In addition to its role in promoting apoptosis, steatosis and fibrosis, ASMase has also emerged as a crucial link in the regulation of methionine metabolism and PC homeostasis, which have been shown to mediate NASH progression. In particular, methionine adenosyltransferases I/III (encoded by MAT1A), the key enzymes involved in methionine metabolism, and ASMase-induced ceramide generation engage in a reciprocal inhibitory process in which increased ceramide levels generated by ASMase repress MAT1A expression, leading to increased ASMase expression, causing a self-sustained vicious cycle of relevance to ASH/NASH development [100,101]. The lower expression of MAT1A from increased ASMase expression results in decreased SAM levels, which are key for the synthesis of PC from phosphatidylethanolamine (PE) via PE methyltransferase (PEMT). Thus, targeting ASMase may boost MAT1A activity and SAM levels, which in turn, may impact the maintenance of an adequate PC to PE ratio to improve NASH. In contrast to ASMase, the role of NSMase in ASH/NASH has been less characterized, and consistent with its controversial function as a mediator of TNF-induced hepatocellular apoptosis, the contribution of NSMase to NASH remains to be fully established [102,103].

### 4.2. Hepatocellular Carcinoma (HCC)

HCC is the end stage of chronic liver disease and the culmination of metabolic diseases, particularly NASH. HCC is the most prevalent form of liver cancer and one of the leading causes of cancer-related deaths in the world. Unfortunately, current therapy is limited and inefficient, and advanced HCCs develop resistance to chemotherapy, making early diagnosis essential for survival. As some lipids such as SLs have been known to mediate cell death pathways, several therapeutic approaches target the regulation of these molecules in order to halt HCC progression. Ceramide levels are markedly reduced in HCC tissues due to the increased expression of strategies that stimulate its degradation, and hence, raising ceramide levels specifically within the tumor may be a relevant therapeutic approach [104]. In this regard, the celecoxib-mediated activation of ER-stress has been shown to induce de novo ceramide biosynthesis, resulting in enhanced apoptosis in hepatoma HepG2 cells [105]. Targeting ACDase, an enzyme that hydrolyzes ceramide (Figure 1), with LCL521 or carmofur has been shown to be of potential relevance in cancer [106,107]. Furthermore, NSMase-1 was reported to be downregulated in HCC tissues [108], and NSMase-2 deficiency promotes liver tumor development by regulating the survival and proliferation of cancer stem-like cells [109]. Despite these findings, many solid tumors, including HCC, develop strategies contributing to the development of chemoresistance. For instance, ceramide-modifying enzymes, particularly glucosylceramide synthase (GCS), are upregulated during sorafenib treatment in hepatoma cells (HepG2 and Hep3B), decreasing ceramide-induced cell death activation and therefore conferring resistance to the sorafenib treatment. In line with these findings, GCS silencing or pharmacological GCS inhibition sensitized hepatoma cells to sorafenib exposure [110].

The contribution of SLs and SMases to HCC has been further suggested to be mediated through their interactions with the mTOR pathway and autophagy, both of which are involved in HCC pathogenesis. The upregulation of the mTORC2 pathway has been shown to promote hepatocarcinogenesis, in part, through the stimulation of de novo fatty acid and lipid synthesis, which leads to steatosis and tumor development [111]. Indeed, increased lipogenesis correlated with elevated mTORC2 activity and human HCC incidence, and the inhibition of fatty acid or SL synthesis prevented tumor development, indicating a link between steatosis and tumor generation and therefore further suggesting that hepatosteatosis acts as a driving force for HCC progression [112]. Although autophagy acts as a mechanism to provide nutrients for energy generation to maintain key cellular functions, autophagy plays a paradoxical role in HCC. For instance, autophagy has been shown to protect cancer cells from the accumulation of damaged organelles and protein aggregates, preventing cell death and the toxicity of cancer therapies [113]. Consequently, the pharmacological or siRNA-mediated inhibition of autophagy has been reported to sensitize HCC cells to the multikinase inhibitor linifanib [114]. However, decreased autophagy markers have been associated with more aggressive HCC phenotypes [115], and several tumor suppressors (e.g., XPD, PTPRO, TAK1 and Klotho) have also been reported to activate autophagy in HCC cells [116,117,118,119]. Among drugs that affect autophagy, vorinostat and sorafenib have been reported to promote carcinoma cell death by increasing ER stress and autophagy via a ASMase/ceramide-dependent pathway [120]. In addition, recombinant human ASMase has emerged as a potential adjuvant treatment with sorafenib in HCC [121]. Moreover, exosomal NSMase-1 has been reported to suppress HCC via decreasing the ratio of SM/ceramide [108]. Overall, although there are emerging data regarding the role of SMases in HCC, further understanding about this link is still needed to optimize the targeting of SLs/SMases as potential therapeutics.

### 4.3. Niemann–Pick Disease Type A/B

Niemann–Pick diseases are a group of autosomal recessive disorders mainly characterized by an excessive accumulation of SLs and cholesterol, primarily in lysosomes. There are two distinct genetic Niemann–Pick disorders. NPA and NPB are caused by a deficient activity of ASMase mostly affecting lysosomal SM homeostasis. Niemann–Pick type C disease (NPC) is characterized by defects in the lysosomal resident proteins NPC1 and NPC2, which are responsible for cholesterol efflux from lysosomes [122,123]. Due to the close association between SLs and cholesterol, both types of lipids accumulate in lysosomes in NPA/B or NPC, regardless of whether the cause is ASMase or NPC1/NPC2 deficiency. Besides SM and cholesterol, other SLs such as gangliosides and sphingosine also accumulate in the lysosomes in NPA/B [73,124,125]. The primary organs affected in NPA patients are the spleen, liver and lung. Consequently, lipid-loaded foam cells are found in a wide variety of organs such as the liver, spleen, lymph nodes, adrenal cortex, lungs and bone marrow, having a severe impact on their correct functioning [126]. NPA and NPB differ in the degree of residual ASMase activity, which determines the impact on clinical features, being more severe in NPA than in type B, and the differential occurrence of neurological symptoms allows diagnosis and prognosis [127]. NPA patients, besides exhibiting hepatosplenomegaly, present a rapidly progressive neurodegeneration with a deep hypotonia, leading to the patient’s death beyond the third year of life [125,128]. NPB patients have no signs of central nervous system involvement, although they may present severe hepatosplenomegaly and liver failure [129,130]. Many NPB patients die before or in early adulthood, often from respiratory or liver failure. An early diagnosis and appropriate handling are thus crucial to lower complication risks, improving quality of life and avoiding extreme procedures such as splenectomy [131,132,133]. Liver biopsy, the diffusion capacity measured by spirometry, the spleen volume and several plasma markers of lipid-laden cells, fibrosis or inflammation are among the biomarkers that are currently used to diagnose NPA and NPB to ensure an appropriate management of the disease [134,135]. Currently, an efficient treatment for NPD type A/B is lacking, and symptomatic therapy has been the only available option for these patients. Enzyme replacement therapy with olipudase alfa, a recombinant form of human ASMase, aims to reduce the non-neurological manifestations of NPD type A/B, showing promising effects in improving liver function and lipid profiles [134]. Although the subsequent increase in ASMase activity following olipudase administration increases ceramide levels, this outcome seems to be transient, and ceramide contents are stabilized after 3 months of treatment [131,132,134,136,137].

ASMase-deficient mice were generated as a mouse model for NPA to study the effects of ASMase deficiency and unravel the molecular pathways involved in this devastating disease [138,139,140,141]. Like NPA patients, ASMase^−/−^ mice present profoundly impaired lipid metabolism and trafficking. The accumulation of lipids in lysosomes is the primary consequence of ASMase deficiency, although this outcome is also observed in other subcellular organelles, including mitochondria, resulting in an overall disruption of cellular homeostasis. Lysosomal SM accumulation has been reported to inhibit the lysosomal transient receptor potential Ca^2+^ channel, impairing endolysosomal trafficking, protein degradation and macroautophagy [142]. In addition, the impaired cholesterol trafficking observed in NPA causes oxidative stress and affects the vesicle trafficking pathways mediated by Rab proteins as well as the fusion of the late endosomal/lysosomal compartments [143,144,145]. Interestingly, despite ASMase deficiency, ASMase^−/−^ mice present elevated ceramide levels in the affected organs, which is possibly due to a breakdown of the accumulated SM in non-lysosomal compartments by other functional SMases. Whether the increase in ceramide levels could contribute to the pathogenesis of NPA remains to be fully investigated [123]. In addition to the ASMase function in lysosomes, some studies have also suggested a role for ASMase in response to stress at the plasma membrane, where it seems to participate in different signaling pathways [146,147]. Thus, the consequences of ASMase deficiency may extend beyond lysosomes and affect other subcellular compartments. In line with this possibility, emerging evidence indicates that lysosomal–mitochondrial interactions are also altered in NPA disease, involving impaired mitophagy due to increased lysosomal cholesterol-mediated impairment of the fusion of autophagosomes containing mitochondria with lysosomes, resulting in mitochondrial dysfunction and overall contributing to disease progression [148,149].

## 5. Role of SMases in Liver Injury and Metabolic Liver Diseases

Besides the aforementioned prevalent chronic liver diseases, SMases have also been related to a wide variety of other liver disorders.

### 5.1. Ischemia–Reperfusion (I/R) Liver Injury

Hepatic ischemia/reperfusion (I/R) injury is a serious complication that compromises liver function because of extensive hepatocellular loss, which impacts diverse clinical settings such as liver surgery and liver transplantation. I/R is caused by the restoration of blood circulation after a period of ischemia in which the supply of oxygen and nutrients is curtailed. I/R results in severe cellular injury, with inflammation and oxidative stress as the main culprits. In the liver, SLs and ceramides, in particular, have been described as signaling lipid intermediates playing a significant role in the stress response and cell death [15,150,151,152]. SMases, as mediators of ceramide production, have also been reported to participate in cell death events and thus play a crucial role in I/R liver injury. ASMase inhibition, either pharmacologically using imipramine or by siRNA-mediated silencing, prevented ceramide increase after hepatic I/R injury and attenuated serum ALT levels, hepatocellular necrosis, cytochrome c release and caspase-3 activation, indicating the relevance of ASMase in this type of liver injury [153]. Apart from ASMase, NSMase has also been linked to I/R, as it has been reported that the inhibition of NSMase decreases the enhanced levels of nitrosative and oxidative stress in I/R injury [154]. Moreover, although NSMase inhibition does not alleviate ER stress, this event downregulates apoptotic stimuli during I/R injury, arguing in favor of a direct role of NSMase in I/R liver injury and the significant protective effect of selective NSMase inhibition for future therapies [155].

### 5.2. Drug-Induced Liver Injury (DILI)

DILI is a major cause of liver failure due to hepatocellular demise upon exposure to a toxic dose of drugs or xenobiotics. Acetaminophen (APAP) hepatotoxicity is the prototype DILI paradigm since APAP is one of the most used pain killers worldwide. Although relatively safe, APAP is a dose-dependent hepatotoxin and a major cause of acute liver failure requiring liver transplantation [156,157]. APAP metabolism generates N-acetyl-p-benzo-quinoenimine (NAPQI), a toxic electrophile, which is detoxified by conjugation with GSH. Excess APAP consumption or a limited hepatic GSH pool favors the binding of NAPQI to mitochondrial protein thiols, leading to the disruption of mitochondrial function and release of generated ROS and oxidative stress. Excess mitochondrial ROS generation potentiates mitochondrial JNK translocation, which amplifies the induction of mitochondrial permeability transition pore opening, leading to further ROS generation, ATP depletion and subsequent hepatocellular death [158,159,160,161,162,163]. As APAP-induced injury mainly impacts mitochondria, the elimination of APAP-induced mitochondrial damage by mitophagy protects against APAP hepatotoxicity.

As indicated above, ASMase deficiency induces the accumulation of SM and other lipids within lysosomal membranes, affecting membrane structure and dynamics. Lysosomal cholesterol accumulation induced by ASMase deficiency decreases mitophagy due to the defective fusion of mitochondrion-containing autophagosomes with lysosomes, resulting in sensitization to APAP hepatotoxicity. Thus, a protective role for ASMase in APAP emerges due to the maintenance of SM/cholesterol homeostasis, which in turn, impacts the turnover of mitochondria and the clearance of defective organelles that sustain APAP-mediated liver injury [148].

### 5.3. Viral Hepatitis B (HBV)

Along with viral hepatitis C, viral hepatitis B (HBV) is a major form of chronic liver disease characterized by severe inflammation and liver injury that can further lead to complications such as cirrhosis, liver failure and liver cancer. The link between SMases and viral hepatitis is still not completely clear, but it is related to the capacity of SLs to control extracellular vesicle (EV) formation. EVs are bilayered particles that carry diverse types of molecules such as proteins, lipids and nucleic acids. Their role as signaling complexes in order to perform intercellular communication has been reported to impact several physiological processes [164,165,166]. SLs, and particularly ceramides, can control the formation of EVs due to the effects on structural and physical properties exerted in lipid membranes [167,168,169]. Ceramides can induce lateral phase separation and domain formation in membrane bilayers and can promote negative curvature in the membrane as well as membrane invagination [170]. Furthermore, lipid phases have been shown to be dependent on ASMase activity. ASMase regulates the structural domains of scaffold molecules, which in lysosomes, have been reported to have a severe impact in death-receptor related liver diseases [171]. EVs play a particular role in viral hepatitis, as these vesicles can transport viral particles, overall enabling the expansion of the viral infection to surrounding cells [172,173]. Specifically, EVs are important for HBV-infected hepatocytes [174]. The production of EVs carrying HBV DNA from HBV-infected hepatocytes has been reported to depend on the ASMase/ceramide system to mediate exosome formation [175], indicating that the regulation of this pathway could be an important therapeutic approach to preventing EV-mediated HBV infection.

### 5.4. Hepatobiliary Diseases

Hepatobiliary diseases comprise a large and heterogeneous group of diseases that affect the hepatic and biliary system. These disorders can be developmental or congenital and can arise at different stages during life. Alk-SMases have been widely studied in the digestion context, as intestinal Alk-SMases have been reported to play an important role in SM digestion, colon cancer prevention and cholesterol absorption [176,177,178]. However, little is known about the role of Alk-SMases in liver diseases. A reduction of Alk-SMase activity was detected in liver specimens from patients with primary sclerosing cholangitis (PSC) [179]. Moreover, a reduction of Alk-SMase activity was also reported in the bile from PSC patients and in patients with cholangiocarcinoma [179,180]. Overall, current evidence suggests that there may be an association between reduced Alk-SMase activity and the progression of hepatobiliary disease. In addition, as Alk-SMases are regulated by bile salts, bile salt diversion was found to strongly reduce Alk-SMase activity in the small intestinal content and feces in rats, indicating that hepatobiliary diseases may also affect intestinal SM digestion [181,182].

### 5.5. Wilson Disease

Wilson disease is an autosomal recessive disorder caused by inactivating mutations in ATP7B, an enzyme involved in the secretion of Cu^2+^ from the liver. The defect results in the accumulation of Cu^2+^ in hepatocytes and other tissues including neuronal, blood or muscle cells [183,184]. An excess of Cu^2+^ ions in cells and tissues induces severe disorders including progressive hepatic cirrhosis, chronic active hepatitis or even progressive hepatic failure, Fanconi syndrome, neurological and psychiatric symptoms, cardiomyopathy, osteomalacia and, in some individuals, anemia. Although Cu^2+^ is an essential trace element in the human diet and is required as a cofactor for the function of diverse proteins, Cu^2+^ triggers the release of ROS that seem to be crucially involved in the induction of cell death by Cu^2+^ [185,186,187]. The release of ROS after cellular treatment with Cu^2+^ has been described as occurring predominantly in lysosomes and mitochondria, and lysosomal membrane damage precedes Cu^2+^ cytotoxicity. Furthermore, Cu^2+^ triggers, at least in erythrocytes, the formation of lipid peroxides and inhibits the activity of antioxidant enzymes, finally resulting in oxidative cell damage, the denaturation of hemoglobin and hemolytic anemia.

Recent studies have shown that Cu^2+^ triggers hepatocyte apoptosis through the activation of ASMase and the release of ceramide [188]. A genetic deficiency of ASMase prevented Cu^2+^-induced hepatocyte apoptosis, while ASMase inhibition with desipramine in rats with a mutation in the Atp7b gene, a genetic model of Wilson’s disease, protects against Cu^2+^ -induced hepatocyte death and liver failure. In line with these findings, individuals with Wilson disease showed elevated plasma levels of ASMase and displayed a constitutive increase in ceramide and phosphatidylserine-positive erythrocytes. The concentrations of free Cu^2+^ (1–3 mM) required to elicit the activation of the ASMase, ceramide release and the apoptosis of hepatocytes are in the range of the concentrations encountered in the plasma of individuals with Wilson disease, thus supporting the clinical significance of the activation of ASMase by Cu^2+^ and its involvement in Wilson´s disease.

## 6. Future Perspectives

Ceramide has been the preferential target of biomedical research deciphering the biological role of SLs. Although considered as critical players in membrane bilayers due to their crucial role in determining their physical properties, SLs and ceramide, in particular, have been recognized as second messengers acting as intermediates of a number of stimuli that regulate multiple cellular functions. While de novo ceramide synthesis is slow but sustained, its generation via NSMase or ASMase represents a fast mechanism for ceramide generation, and hence, both SMases stand as efficient molecular devices for releasing ceramide almost instantly in response to different stimuli, which lends further support to positioning both enzymes as key intermediates in signaling pathways. A common feature of NSMase and ASMase is that both hydrolyze SM embedded in membrane bilayers. However, their pH dependence marks an important difference between the enzymes, and accounts for the generation of discrete pools of ceramide in specific cellular sites. In this regard, NSMase requires a neutral pH for optimal activity and generates ceramide in the vicinity of the inner leaflet of the plasma membrane, with NSMase-2 being of major relevance in pathophysiology. On the other hand, ASMase, which requires an acid pH for activity, is located in acidic compartments, predominantly in lysosomes, where it hydrolyzes lysosomal SM pools. Quite interestingly, lysosomal ASMase processing at the Golgi or following its exocytosis generates a secretory form of ASMase, which acts at the plasma membrane to generate ceramide from local SM hydrolysis. Since the molecular identity of ceramide is determined by the length of the fatty-acyl chain linked to the carbon backbone, the hydrolysis of SM at different membrane bilayers by individual SMases contributes to the generation of unique molecular species of ceramide. Thus, the activation of either NSMase or ASMase represents a specific mechanism for the generation of different ceramide species that likely accounts for their diverse biological effects. Although in physiological settings, the generation of ceramide occurs predominantly by de novo synthesis, the activation of SMases in response to different triggers, such as inflammatory cytokines, stress or chemotherapy, results in the rapid generation of ceramide in specific, discrete cellular sites that targets particular pathways involved in chronic liver diseases. Of particular relevance is the role of SMases in NASH and its progression to HCC, which has escalated to become the most important chronic liver disease worldwide and is expected to increase in the near future due to its association with obesity and the type 2 diabetes epidemic. Thus, elucidating the mechanisms of activation and identification of intermediates involved in SMase signaling may be of relevance for the treatment of prevalent liver diseases. As a proof of concept, targeting ASMase with tricyclic antidepressants, such as amitriptyline, has been initially reported in preclinical models of both ASH and NASH. Since the complete inhibition of ASMase may result in serious complications as disclosed by the deletion of ASMase in NPA, the exposure time and dose for ASMase inhibitors need to be carefully chosen to allow residual ASMase activity to avoid an undesirable NPA phenotype, a further complication that may not be relevant for NSMase. Further research would be required to develop specific and reversible SMase inhibitors as potential treatments for liver diseases.

## Figures and Tables

**Figure 1 biomolecules-10-01497-f001:**
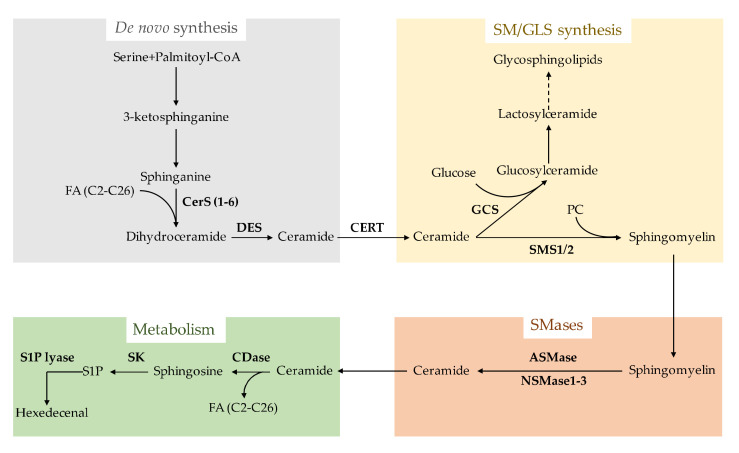
Synthesis and metabolism of sphingolipids (SLs). Ceramide is the prototype SL, which is synthesized de novo in the ER from serine and palmitoyl-CoA (upper left panel). The molecular identity of ceramide is determined by the length of the acyl chain linked to the carbon backbone. Six different ceramide synthases (CerS 1–6) exhibit differential affinity towards fatty acids of different length (C2–C34). Once synthesized in the ER, ceramide is transported to the Golgi (upper right panel) and serves as the substrate for glucosylceramide synthase (GCS) to generate glucosylceramide or sphingomyelin synthases (SMS1/2) to yield sphingomyelin from phosphatidylcholine (PC). The distribution and subsequent hydrolysis of sphingomyelin in different membrane bilayers by SMases (lower right panel) represents a fast mechanism of almost instant ceramide generation. Ceramide can be catabolized by ceramidase (CDase) (lower left panel) to generate sphingosine. Sphingosine can be phosphorylated by sphingosine kinase (SK) into sphingosine 1-phosphate (S1P), a bioactive lipid, which can be further degraded by S1P lyase into hexadecenal. In addition, the pool of sphingosine generated by CDase can be reacylated by CerS back into ceramide in the so-called salvage pathway.

**Figure 2 biomolecules-10-01497-f002:**
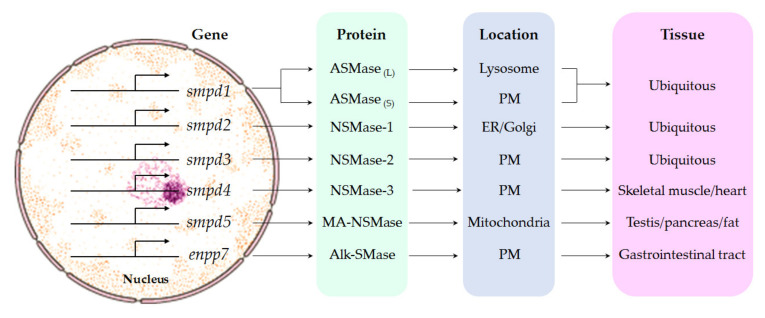
Types and characteristics of mammalian sphingomyelinases (SMases). SMases are encoded by different genes (smp1–5; enpp7), which results in 7 different proteins: two acid sphingomyelinases (ASMases), four neutral sphingomyelinases (NSMases), including the mitochondrial-associated NSMase (MA-NSMase) and alkaline sphingomyelinase (Alk-SMase). Please note that the lysosomal ASMase (ASMase _L_) and the secretory ASMase (ASMase _S_) are encoded by smpd1 and localized in different membrane bilayers, namely, lysosomes and the plasma membrane (PM), respectively. ASMase and NSMase differ in their optimal pH for maximal activity and requirement for specific cations for activation and exhibit differential distribution within the cell and in specific tissues.

**Figure 3 biomolecules-10-01497-f003:**
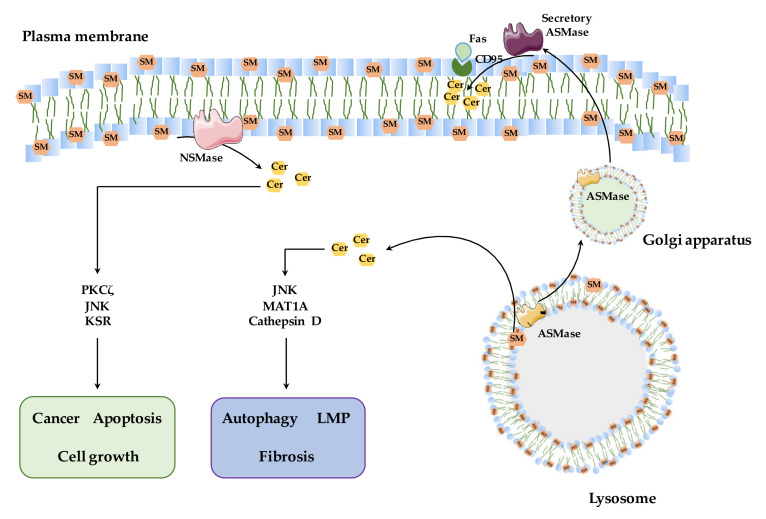
Functional role of NSMase and ASMase in signaling pathways. NSMase and ASMase are localized in different membrane bilayers, where they hydrolyze specific sphingomyelin (SM) pools, mostly in the plasma membrane and lysosomes, respectively, consistent with their pH optima for activity. NSMase-induced ceramide generation in the vicinity of the plasma membrane activates specific targets, e.g., PKCd, KSR or JNK, and is mainly involved in cancer, apoptosis and cell growth. ASMase, on the other hand, hydrolyzes lysosomal SM, and its deficiency causes Niemann–Pick type A (NPA) disease, a lysosomal storage disorder characterized by the accumulation of SM in lysosomes. Ceramide generated by ASMase activation has been shown to target MAT1A and cathepsin D as well as JNK and is involved in the regulation of autophagy, hepatic fibrosis and lysosomal membrane permeabilization (LMP). A subset of ASMase traffics to the Golgi and is secreted to the plasma membrane. The secretory ASMase hydrolyzes SM at the outer leaflet of the plasma membrane, and the resulting ceramide causes the death activation receptors, i.e., CD95, to bind Fas ligand. This pool of ceramide mediates Fas-induced liver injury and failure.

**Table 1 biomolecules-10-01497-t001:** Major highlights of the role of NSMase and ASMase in liver diseases.

Disease	Protein	Function
Alcoholic and non-alcoholic steatohepatitis (ASH/NASH)	ASMase	Triggers hepatocellular apoptosis in response to TNF and Fas-induced fulminant liver injury. Is required in alcohol or HFD-induced lipogenesis and macrosteatosis. Is required for ER stress (either alcohol or HFD-induced or autophagy suppression-mediated). Is activated during HSC activation and required for their transdifferentiation to myofibroblast-like cells that promote fibrogenesis. Is a crucial link in the regulation of methionine metabolism and PC homeostasis mediating NASH progression.
	NSMase	Less characterized in ASH/NASH. Controversial function of TNF-induced hepatocellular apoptosis.
Hepatocellular carcinoma (HCC)	NSMase-1	Is downregulated in HCC tissues
	NSMase-2	Its deficiency promotes liver tumor development by regulating the survival and proliferation of cancer stem-like cells
	ASMase	Promotes cell death by increasing ER stress and autophagy
Niemann–Pick A/B (NPA/B)	ASMase	Its deficiency affects lysosomal sphingolipid accumulation, resulting in lipid-loaded foam cells in a wide variety of organs having a severe impact in their correct functioning. Its deficiency impairs cholesterol trafficking causing oxidative stress and affects vesicle trafficking pathways mediated by Rab proteins as well as fusion of the late endosomal/lysosomal compartments Its deficiency alters lysosomal–mitochondrial interactions, involving impaired mitophagy, resulting in mitochondrial dysfunction and overall contributing to disease progression.
Ischemia–reperfusion (I/R) liver injury	ASMase	Its inhibition prevents ceramide increase after hepatic I/R injury, attenuating serum ALT levels, hepatocellular necrosis, cytochrome c release and caspase 3 activation.
	NSMase	Its inhibition decreases enhanced levels of nitrosative and oxidative stress in I/R injury. Its inhibition downregulates apoptotic stimuli during I/R injury
Drug-induced liver injury (DILI)	ASMase	Its deficiency alters lysosomal–mitochondrial interactions, involving impaired mitophagy, resulting in mitochondrial dysfunction and sensitization to APAP hepatotoxicity.
Viral hepatitis B (HBV)	ASMase	Is required for the production of HBV-DNA carrying extracellular vesicles (EV), essential for hepatocyte infection.
Hepatobiliary diseases	Alk-SMase	Its activity is reduced in the bile and liver from primary sclerosing cholangitis (PSC) patients Bile salt diversion strongly reduces Alk-SMase activity in the small intestinal content, which may also affect intestinal SM digestion
Wilson disease	ASMase	Cu^2+^ triggers hepatocyte apoptosis through activation of ASMase and the release of ceramide

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
