# Peer review of "Sphingomyelinases and Liver Diseases"

_biomolecules, 2020, doi:10.3390/biom10111497_

Round 1
Reviewer 1 Report
Many thanks for asking me to review this very comprehensive review of sphingomyelinases and liver disease.
The review is very thorough and well written throughout with clear aims and layout. The referenced literature is very comprehensive and up to date. I have few minor comments which I feel should be addressed:
- The manuscript is very pre-clinical and offers little translational insight. Whilst the authors briefly discuss this at the very end, there is little mention of clinical agents that are able to target the ceramics pathway. There are a number of small molecular inhibitors (eg. LCL 251) along with chemotherapeutic agents (eg. carmofur) that are able to inhibit acid ceramidase. Could the authors provide greater insight into how this pathway might be targeted in the clinical setting.
- A single table summarising pre-clinical and clinical models in each of the disease would be really helpful for readers.
- Could the authors provide a brief summary of the burden of each of the diseases they describe ie. symptoms caused, life expectancy after diagnosis to ensure there is some context as to the urgency of improving treatments in these areas.
- The diagrams summarising the pathways would benefit from the addition of which parts of the pathway may be amenable to molecular/pharmacological manipulation.
Author Response
1.The manuscript is very pre-clinical and offers little translational insight. Whilst the authors briefly discuss this at the very end, there is little mention of clinical agents that are able to target the ceramics pathway. There are a number of small molecular inhibitors (eg. LCL 251) along with chemotherapeutic agents (eg. carmofur) that are able to inhibit acid ceramidase. Could the authors provide greater insight into how this pathway might be targeted in the clinical setting.
We want to thank the reviewer for the comments and suggestions to highlight the potential relevance that the regulation of ceramide may have in translational hepatology. We agree that the manuscript focused on pre-clinical observations, and although much progress has been made in understanding the role of sphingolipids and ceramide in liver pathology, the application of this knowledge to clinical practive is still in its infancy, indicating that further basic research is required to sustain the targeting of ceramide or SMases in chronic liver diseases.
Nevertheless, we feel that the manuscript contains reference in different sections to the potential role of inhibiting SMases, particularly ASMase in ASH and NASH and fibrosis. In addition, we have followed the suggestion of the reviewer and have made reference to the potential relevance of inhibition of ACDase by LCL 251 or carmofur as adjuvants for cancer treatment in page 9 of the revised version.
- A single table summarising pre-clinical and clinical models in each of the disease would be really helpful for readers.
We have followed this suggestion and have included a table summarizing the experimental models used for the different liver diseases.
- Could the authors provide a brief summary of the burden of each of the diseases they describe ie. symptoms caused, life expectancy after diagnosis to ensure there is some context as to the urgency of improving treatments in these areas.
Although we agree and appreciate this suggestion, we feel that this may be out of the scope of the present review as these additions would be too clinicial in nature and perhaps not very relevant for the focus of the review.
- The diagrams summarising the pathways would benefit from the addition of which parts of the pathway may be amenable to molecular/pharmacological manipulation.
Since the main focus of the manuscript is to document and summarize the role of SMases in liver diseases, whose targeting are covered within the text, showing other pathways amenable for molecular/pharmacological manipulation may be out of the scope of the review since these pathways are not directly linked to the action and role of SMases in liver disease.
Reviewer 2 Report
This review is dedicated to sphingomyelinases, enzymes that hydrolyze sphingomyelinas by producing ceramide. It describes their physiological activity and their role in liver diseases. The review is mostly comprehensive and clear.
The excessive use of acronyms makes reading difficult and some acronyms are not even explained (e.g. CERT, SH). I suggest reducing the number of acronyms and/or adding a table or list to help the reader.
There are two points which in my opinion are wrong or not sufficiently explained.
- Why the authors insist on zinc ion dependence of the secreted form of ASMase? (lines 118-122, 569-573) The endosomal form is encoded by the same gene, so presumably this one too has amino acids that coordinate zinc and needs zinc. In fact, the structure of the protein, published in 2016, reports the presence of two zinc ions (PMID 27435900).
- If SMase, as the authors of the review also say, is secreted, it acts on the outside of the plasma membrane. Why in Figure 3 the protein is represented on the inner leaflet? Recently (PMID 31155842) an alternative explanation (to the traffic through the Golgi) for the presence of ASMase on the cell surface has also been given, the authors should describe also this possible route.
Minors:
Line 60, please reformulate the phrase: ‘the catabolism of ceramide into sphingosine can be reacylated to generate ceramide ’
Line 70 ‘As SMases represent the predominant pathway for the sudden generation of ceramide in the present review we summarize the role of SMases in liver pathology’ this motivation does not hold up, first you should mention the importance of ceramide in liver diseases.
Line 129: I think there is a preposition missing in the following sentence (Among the) ‘four different mammalian NSMases characterized so far’
Author Response
Reviewer 2:
1.Why the authors insist on zinc ion dependence of the secreted form of ASMase? (lines 118-122, 569-573) The endosomal form is encoded by the same gene, so presumably this one too has amino acids that coordinate zinc and needs zinc. In fact, the structure of the protein, published in 2016, reports the presence of two zinc ions (PMID 27435900).
We thank the reviewer for pointing out this important point to us. The regulation and function of the lysosomal and secretory ASMases are still not fully understood and this includes their differential dependence of the Zn2+ for activity. We agree with the reviewer´s comments and have removed the statements for the role of the Zn2+ dependence on the secretory ASMase, as suggested, and have incorporated the indicated observations by the reviewer, as shown in page 4 of the revised version, quoting the two references.
2.If SMase, as the authors of the review also say, is secreted, it acts on the outside of the plasma membrane. Why in Figure 3 the protein is represented on the inner leaflet? Recently (PMID 31155842) an alternative explanation (to the traffic through the Golgi) for the presence of ASMase on the cell surface has also been given, the authors should describe also this possible route.
We agree with the reviewer´s comments and have modified Figure 3 to denote this aspect showing that the secretory ASMase acts on outer leaflet of the bilayer. In addition, the suggested observations pointing to an alternative mechanism for the genesis of secretory ASMase are very interesting and have included this aspect in the revised version, in page 4 of the manuscript.
Minor
Line 60, please reformulate the phrase: ‘the catabolism of ceramide into sphingosine can be reacylated to generate ceramide ’
We have rephrased this sentence as requested by the reviewer, as shown in page 3 of the revised manuscript.
Line 70 ‘As SMases represent the predominant pathway for the sudden generation of ceramide in the present review we summarize the role of SMases in liver pathology’ this motivation does not hold up, first you should mention the importance of ceramide in liver diseases.
We have followed this suggestion and have modified the sentence as requested, in page 3 of the revised version.
Line 129: I think there is a preposition missing in the following sentence (Among the) ‘four different mammalian NSMases characterized so far’
We apologize for this mistake, which has been corrected in the revised version.